behaviour/ecology/evolution

central-place foraging, hunter–gatherer mobility, social networks, movement ecology, human behavioural ecology, agent-based model

**Authors for correspondence:**
Ketika Garg
e-mail: ketikagarg@gmail.com
Cecilia Padilla-Iglesias
e-mail: cecilia.padillaiglesias@uzh.ch

†Contributed equally.

# Hunter–gatherer foraging networks promote information transmission

Ketika Garg[1,†], Cecilia Padilla-Iglesias[2,†], Nicolás Restrepo Ochoa[3] and V. Bleu Knight[4]

[1]Department of Cognitive and Information Sciences, University of California, Merced, CA, USA
[2]Institute of Anthropology, University of Zurich, Zurich, Switzerland
[3]Department of Sociology, Duke University, Durham, NC, USA
[4]Active Inference Lab, USA

KG, 0000-0003-0915-7314; CP, 0000-0003-1302-5955

Central-place foraging (CPF), where foragers return to a central location (or home), is a key feature of hunter–gatherer social organization. CPF could have significantly changed hunter–gatherers' spatial use and mobility, altered social networks and increased opportunities for information-exchange. We evaluated whether CPF patterns facilitate information-transmission and considered the potential roles of environmental conditions, mobility strategies and population sizes. We built an agent-based model of CPF where agents moved according to a simple optimal foraging rule, and could encounter other agents as they moved across the environment. They either foraged close to their home within a given radius or moved the location of their home to new areas. We analysed the interaction networks arising under different conditions and found that, at intermediate levels of environmental heterogeneity and mobility, CPF increased global and local network efficiencies as well as the rate of contagion-based information-transmission. We also found that central-place mobility strategies can further improve information transmission in larger populations. Our findings suggest that the combination of foraging and movement strategies, as well as the environmental conditions that characterized early human societies, may have been a crucial precursor in our species' unique capacity to innovate, accumulate and rely on complex culture.

## 1. Introduction

One of the pivotal transitions in human evolution is our ability to generate, accumulate and rely on complex, cumulative culture [1–4]. Recent evidence from hunter–gatherer societies [5,6] suggests that changes in our ancestors' social networks and connectivity could have promoted such a transition by facilitating an efficient exchange

and transmission of cultural information. Given that the frequency and nature of social interactions between hunter–gatherers would have been affected by their movement and spatial distribution patterns, researchers have proposed that divergences in foraging behaviour, coupled with ecological changes, could have led to changes in the dynamics of social interactions and hence patterns of social organization [7–9]. However, the impact of hunter–gatherer foraging and movement behaviour on emergent social networks and their ability to transmit information is still not thoroughly understood (but see [10,11]).

Central-place foraging (CPF) marks a critical behavioural change between the foraging styles of early hominins and our closest great ape relatives that would have modified their movement and consequently spatial and social patterning [12–14]. Non-human Great Apes (henceforth Great Apes) tend to consume food when they find it ('point-to-point' foraging), make sleeping nests at variable locations and have short foraging trips [15]. On the other hand, hunter–gatherers establish residential camps or central places around which they systematically forage and bring the food they collect during foraging trips (or logistical forays) back to their camps to share and process it with camp members ('central-place' foraging) [16,17]. In addition, human foragers can make longer foraging trips and periodically move the location of their residential camps to access new resource areas with little overlap in the foraging radii between successive camps. These properties result in an expansion of their overall home ranges during their lifetimes [15] compared with other primates who spend most of their adult lives within the same area, leading to a more restricted use of space [18–22].

Such differences in mobility could have altered spatial patterns and dynamics of social interactions and led to more complex social structures. In particular, we hypothesize that CPF could have played an essential role in the subsequent development of multi-level sociality. In multi-level organizations, sets of multiple core units (such as nuclear families) repeatedly coalesce, intermix and disperse, giving rise to relatively fluid local bands that are embedded in higher-level interconnected regional networks [14,23–27]. These extended, flexible and fluid social landscapes would have increased the likelihood of interactions, and consequently opportunities for social learning, and information exchange compared with the rest of the Great Apes [2,28].

However, hunter–gatherer foraging and mobility decisions (e.g. daily trips, residential movements) are influenced by their resource environments as well as various costs (e.g. travelling costs) and benefits (e.g. resource abundance) of their foraging activities [29,30]. For example, in rich environments where there are plenty of resources available within their foraging radii (or home ranges), hunter–gatherer bands may be able to afford greater sedentism [31]. By contrast, unproductive landscapes may require bands to move their camps multiple times a year due to resource depletion within their foraging radii [32]. In addition, if resources are homogeneously distributed, ethnographic studies have shown that bands tend to predominantly rely on short and frequent residential moves [17]. In these settings, the frequency of encounter with other bands and thus, the interconnectivity between them could decrease. Conversely, if resources are heterogeneously distributed and some areas are more resource-rich than others, bands may aggregate in key locations from which they conduct daily trips to procure resources [29], potentially generating more opportunities for interactions [33].

In this paper, we model and compare point-to-point and CPF (with different home-range radii) behaviour across a range of environments. We investigate the effect of CPF and mobility on the interaction patterns between foraging units and the subsequent social networks that are formed due to foraging units coinciding on resources. We then test the efficiency of information transmission in the networks that emerge from the different mobility regimes and environments. Previous theoretical and computational models have explored the effects of environmental heterogeneity on social networks emerging from foraging behaviour across different environments [34] and hunter–gatherer mobility on cultural transmission [10,35]. However, models explicitly linking foraging strategies, environmental features and hunter–gatherer interaction networks remain lacking. Our work illustrates a direct connection between environmental conditions, foraging behaviour and information flow in hunter–gatherer social networks, thereby providing insights into the evolutionary origins of our species' unique ability to innovate, accumulate and rely on complex culture.

## 2. Methods

### 2.1. Model description

We investigated how CPF behaviour would affect the emergent interaction networks across environments. Previous work by Ramos-Fernández *et al.* [34] modelled the effect of environmental

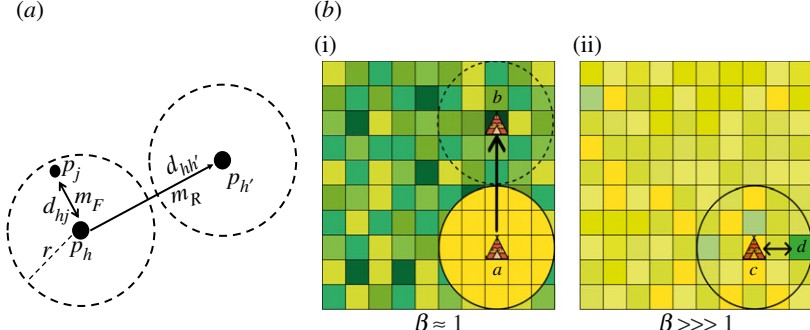

**Figure 1.** Model description. (*a*) A schematic of agent movement in the model. Agents can make foraging movements ($m_F$) within a radius $r$ from their home ($p_h$) to a new patch ($p_j$) or residential movements ($m_R$) to a new home ($p_{h'}$). (*b*) An illustration of the variations in resource environments modulated by the parameter $\beta$. A low value of $\beta$ results in a rich (dark-green patches) and heterogeneous environment (left), whereas a very high value of $\beta$ results in a scarce (yellow patches) and homogeneous environment (right). When food depletes within an agent's radius (yellow patches), it moves its residence ($a \rightarrow b$). Otherwise, it continues to forage within its radius ($c \leftrightarrow d$).

heterogeneity on the interaction networks that emerge from multiple agents foraging independently (representing spider monkeys). The authors showed that a complex social structure with fission–fusion properties, resembling those observed in field studies among real spider monkey societies, could emerge simply from optimal foraging rules in heterogeneous environments.

Our model (henceforth central-place model), like the model from Ramos-Fernández *et al.* [34] (henceforth point-to-point model), was executed in a two-dimensional environment spatially ranging from 0 to 1, and comprising 50 000 uniformly distributed patches. Each patch was initially assigned resource content, $k_i \geq 1$, drawn from a normalized power-law probability distribution, $P(k) \approx Ck^{-\beta}$ where the exponent $\beta$ determined the distribution of resource content and the total resource abundance (see electronic supplementary material, Methods section and figure S1), and $C = 1/\sum_{k=1}^{\infty} k^{-\beta}$ was the normalization constant. Following this equation, the richness of an environment (or abundance) and its heterogeneity (or distribution) co-varied and were determined by $\beta$.

When $\beta \approx 1$, $k$ had a broad range with high values, patches varied widely in their resource content, and the environment was abundant with many rich patches. Conversely, $\beta \gg 1$ corresponded to smaller values and a restricted range of $k$ that resulted in an environment composed of scarcer resources that are homogeneously distributed across patches. Patches were depleted by a unit in their resource content ($k$) every time-step that a foraging unit spent at it, and they did not regenerate (see electronic supplementary material, figure S5 for more information on resource depletion).

Each agent in our simulation represented a monogamous, nuclear family/foraging unit (adult male, female and dependent offspring) which are the core, indivisible units of social organization across hunter–gatherer societies [36]. Since ties between individuals from different families would result in a tie between the families, agents were assumed to forage and move as a single foraging unit. The model was initialized with agents independently and randomly distributed across the patches. Foraging units followed a rule whereby they move to a new patch ($p_j$) from a depleted patch ($p_i$) such that it minimized the cost/gain ratio ($d_{ij}/k_j$), where $d_{ij}$ is the distance between the patches and $k_j$ is the resource content of $p_j$. Our model (figure 1*a*) modified this resource-maximization rule to implement CPF and distinguished between foraging (or logistical) and residential moves [16,17].

In our model, foraging units moved to fixed home locations from which they exploited the surrounding local environment in their foraging radius before moving to another home location. Every foraging unit had a complete knowledge of resources, a randomly allocated home location (i.e. central place), and a foraging area with a given radius, $r$. Foraging units could forage and change their home location based on the following rules: when foraging units were on a patch with no food left, they made foraging moves ($m_F$) to a patch ($p_j$) within $r$ such that the cost/gain ratio ($d_{hj}/k_j$) was minimized, where $d_{hj}$ is distance from the current home ($p_h$), and $k_j$ is the resource content of $p_j$ (figure 1*b*(ii)). Before every move, foraging units compared the cost/gain ratio of patches outside the radius to patches within the radius. When the resource quality within $r$ diminished compared with the rest of the environment (figure 1*b*(i)), instead of making a foraging move to profitable patches outside their radius, foraging units made a residential move. Residential moves ($m_R$) allowed foraging units to select a new home ($p_{h'}$) that minimized ($d_{hh'}/k_{h'}$) but was far enough from the

current base ($d_{hh'} \geq 2^*r$) to avoid overlap [10]. Each time-step that a foraging unit coincided with another foraging unit on a patch, they formed a social network tie or added a unit of weight to an existing tie. This can therefore be considered to represent a tie between two core family units. In the literature, hunter–gatherer family units have been well-documented to regularly interact, fuse and disperse to form a higher level of organization (bands or camps), which our model simulates [5,6,17,18]. At the same time, such bands have been shown to share the same home bases (or central places) and often co-reside. In our model, when two family units coincide on a home patch, they share the same home range and forage together. By contrast, if two foraging units share a foraging patch (as opposed to a home patch), they will have overlapping home ranges which can also increase the likelihood of interactions and result in larger communities [31,37].

To assess how the combination of environmental heterogeneity and CPF strategies affect the emergent social networks, we varied the resource exponent $\beta$ parameter to take values between 1.5 and 4.5 and the foraging radius $r$ to assume values of 1, 0.1, 0.01 and 0.001. We ran the point-to-point model by setting the radius to 0 after verifying this was equivalent to the framework from Ramos-Fernández et al. [34]. We also tested the effect of population size by running the model with 50, 100 and 200 foraging units (electronic supplementary material, figure S1). While our main motivation with this manipulation was to explore the effect of changing population sizes on our model results, the values we chose are ethnographically meaningful. We mostly focused on population sizes of 100 foraging units/families (that correspond to 500 individuals) which have been widely documented [38] and are assumed to represent the average size of hunter–gatherer regional bands or groups [18]. Populations of size 200 correspond to prevalent estimates of the size of entire ethnic populations (or metapopulations) (as compiled in Lehmann et al. [39]). Finally, populations of 50 families (200–250 individuals) represent a lower limit for hunter–gatherer populations to remain viable [40]. We ran 50 simulations for each parameter combination and the point-to-point model and extracted the weighted social networks formed throughout the simulation as well as at the end of the simulation. To ensure that our results are not an artefact of the chosen time-steps, we conducted sensitivity analyses, running each parameter combination for 1000 time-steps (available in the electronic supplementary material). We found the results to be consistent over longer time-steps and thus, report the results from the first 100 time-steps in the following text.

## 2.2. Networks

We extracted the final networks formed from the sum of all interactions by the end of each run. We provide complete summary statistics of each parameter combination's resulting networks in the electronic supplementary material, tables S4–S5. As a robustness check, we also looked at how the networks developed across the simulations. For 100 time-steps, we examined the networks after each interval of 10 time-steps. For 1000 time-steps, we increased that window to a 100 time-steps.

### 2.2.1. Efficiency

We tested the networks for their ability to transmit information by measuring their global and local efficiencies [41]. The efficiency measures have been used across various studies to investigate the transmission of social and cultural information in various networks, including hunter–gatherer social networks [5]. Global efficiency indicates a network's ability to transmit information across the entire network and is inversely related to the characteristic path length (or the average distance between nodes). Latora & Marchiori [41] define a graph's global efficiency as the inverse of the sum of the shortest paths between all nodes $i$ and $j$:

$$E = \frac{1}{n} \sum_{i \in N} \frac{\sum_{j \in N, j \neq i} d_{ij}^{-1}}{n-1}, \tag{2.1}$$

where $N$ is the set of all nodes in the network, $n$ is the total number of nodes and $d$ is the shortest distance between two nodes. On the other hand, local efficiency relates to the clustering coefficient of a network (i.e. the degree to which a node's local neighbourhood is interconnected). It measures the average global efficiency of subgraphs and denotes how well each local neighbourhood can exchange information within itself. We modified the efficiency measures to incorporate weights (for a more detailed description, see electronic supplementary material).

### 2.2.2. Contagion simulations

We conducted contagion simulations on the extracted networks and calculated the proportion of agents that acquired the diffusing information. Each simulation consisted of 5000 time-steps and we conducted 50 simulations on each network to account for variation across runs. For each simulation, we randomly chose one agent and seeded it with the behaviour to be transmitted. As the simulation proceeded, other agents adopted the behaviour with a probability proportional to the number of their neighbours that had already acquired it and the strength of their ties with them. However, the way this probability was calculated varied for simple and complex contagion, and we explored both conditions in our analyses. Simple contagion models a transmission where the probability with which an agent $(Pr(A)_i)$ acquires information is dependent upon the strength of ties and number of neighbours with the information. This can be expressed as

$$Pr(A)_i^{t+1} = \frac{\sum w_i \times d^t}{\sum w_i},$$

where $w_i$ is a vector of the edge weights that $i$ shares with its neighbours and $d$ is a vector of same length containing 1 if the corresponding neighbour has acquired the information or 0 otherwise at time $t$.

To illustrate, consider that the agent $A_i$ has three neighbours $(A_j, A_k, A_l)$ and the ties with them have different weights (2, 5, 10). If $A_l$ has information, then the probability of $A_i$ acquiring it $(P(A_i) \approx 0.59)$ is higher than if $A_j$ had information $(P(A_i) \approx 0.12)$.

We also modelled a more restricted mode of transmission (complex contagion) that increases the dependence on how many of an agent's neighbours have the information and the strength of their ties. Complex contagion is more suited to capture the diffusion of costly or difficult behaviours that are socially acquired (such as cultural traits) and need reiterated affirmation [42,43]. Here, the probability of acquisition now rises exponentially as more neighbours acquire the information

$$Pr(A)_i^{t+1} = \left( \frac{\sum w_i \times d^t}{\sum w_i} \right)^2.$$

Given this type of contagion, the probability of $A_i$ acquiring information from $A_l$ reduces to $P(A_i) \approx 0.34$ and thus, stronger ties and more neighbours with the information would be required to increase the likelihood of transmission.

## 3. Results

### 3.1. Environmental factors affect the efficiency of information transmission in networks

In line with the results from Ramos-Fernández *et al.* [34] (figure 2), we found that environmental heterogeneity strongly influenced the networks formed, with $\beta = 2.5$ generating the most efficient networks ($\overline{E}_{\text{global}} = 0.13$, $\overline{E}_{\text{local}} = 0.65$). In environments of $\beta \approx 1$, where many rich resource patches were available, foraging units had very low mobility (see next section for mobility results) and stayed fixed at a rich resource patch for long durations. In the homogeneous environment of $\beta = 4.5$, every patch had low resource value, and foraging units depleted patches quickly. They frequently moved across the environment resulting in low interaction rates (as evidenced by density of connections) with other foraging units. However, at intermediate heterogeneity and resource abundance ($\beta = 2.5$), foraging units coincided at many different rich patches available in the environment and formed stronger social ties. This can be evidenced by the high number of total interactions between foraging units per time-step (electronic supplementary material, figure S4) that increased the network's local efficiency. On the other hand, an intermediate number of rich patches also enabled more movement and unique interactions between the foraging units that made the network more expansive and increased its global efficiency (electronic supplementary material, figures S12–13). Higher population sizes further increased the rate of interactions between the foraging units and thus, the network efficiencies (electronic supplementary material, figure S1).

### 3.2. Central-place foraging increases global and local network efficiency

We found that point-to-point foraging created networks comprised of isolated foraging units with very high local efficiency (or clustering) but low global efficiency. These networks contained strongly

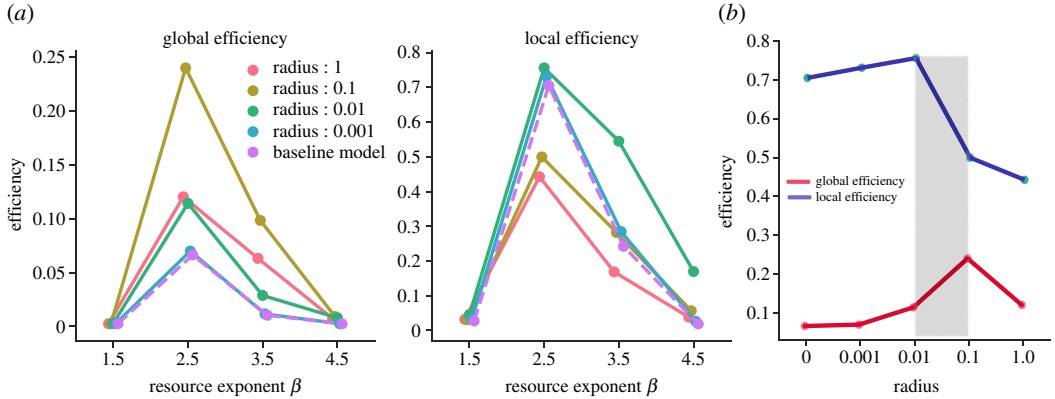

**Figure 2.** Network efficiencies after 100 time-steps. (*a*) The plot shows the average global (left) and local (right) efficiencies of the networks as a function of environmental heterogeneity for each radius. (*b*) The plot shows the relationship between efficiencies and radius for $\beta = 2.5$. Shaded region corresponds to intermediate radii that balances global and local efficiencies.

connected small subgroups of foraging units that were distributed across the environment with few to no connections between them. By contrast, CPF increased the number of unique interactions between foraging units (electronic supplementary material, figure S12–13) and created ties between otherwise unconnected subgroups (or components) that resulted in a more interconnected and expansive network (electronic supplementary material, figure S7). A completely disconnected graph has a number of components equal to its nodes, while a fully connected graph has a single component. We found that the ties between subgroups decreased the number of components and increased networks' global efficiency while maintaining high local efficiency. On the one hand, this formed strongly bonded local groups, and on the other hand, large-scale, interconnected regional networks such as the ones observed among ethnographic hunter–gatherers [2,5,25,44].

To explore the effect of different radii of CPF, we compared the different mobility regimes (frequency and magnitude of residential and foraging moves) across environments and radii (figure 4; electronic supplementary material, tables S2–S3).

When the foraging radius was small ($r = 0.001$), at intermediate levels of environmental heterogeneity ($\beta = 2.5$), the observed foraging pattern closely resembled point-to-point foraging [15], where foraging units do not return to a central place, make short residential moves ($\overline{n}_r = 26, \overline{d}_r = 0.022$) and fewer and shorter foraging moves ($\overline{n}_f = 3, \overline{d}_f = 0.001$). The resulting networks from $r = 0.001$ comprised many (approx. 24) densely connected subgroups of high local efficiencies ($\overline{E}_{local} = 0.73$) (electronic supplementary material, table S4). Nonetheless, these dense subgroups lacked connections between them with a maximum of three subgroups connected to each other (electronic supplementary material, table S5) (figure 3).

Increasing the foraging radius to intermediate values ($r = 0.01$ and $r = 0.1$) resulted in foraging units making longer and more frequent foraging moves combined with longer but fewer residential moves. At $r = 0.01$, a small increase in residential mobility ($\overline{n}_r = 16, \overline{d}_r = 0.048$) created a few longer connections between the dense subgroups. These connections resulted in more subgroups being connected (4–5) and a more expansive network that increased the global efficiency ($\overline{E}_{global} = 0.11$). However, the network still remained highly cliquish with high local efficiency ($\overline{E}_{local} = 0.75$).

As the foraging radius and use of space further increased ($r = 0.1$), foraging units moved less frequently ($\overline{n}_f = 20, \overline{n}_r = 2$) but undertook longer moves ($\overline{d}_f = 0.03, \overline{d}_r = 0.23$). This change in mobility increased the long-range connections between fewer (approx. 15) but larger and interconnected subgroups (6–8). The resultant subgroup structure made the network substantially more efficient at the global scale ($\overline{E}_{global} = 0.24$) while maintaining considerable local efficiencies ($\overline{E}_{local} = 0.50$) (electronic supplementary material, tables S4–S5). However, we found that both efficiencies decreased compared with intermediate radii ($\overline{E}_{local} = 0.44, \overline{E}_{global} = 0.12$) when the foraging units had a very large foraging radius ($r = 1$). In the absence of residential moves, the foraging units remained tethered to their original home and traversed longer foraging moves ($\overline{d}_f = 0.05$) to find food. The longer moves helped create long-range connections between foraging units that resulted in a large number of connected subgroups (7–11) and a more globally efficient network than the point-to-point model ($\overline{E}_{global} = 0.07$). Nonetheless, the strong tethering decreased the overall use of space and the probability of coinciding with others for longer durations, resulting in fewer and weaker connections between subgroups (see electronic supplementary material, figure S2–S3) with low local efficiencies.

**Figure 3.** Emergent networks from 100 foraging units after 100 time-steps. The plot shows an example of weighted networks that emerge from different foraging behaviours in $\beta = 2.5$ environment (from (a) to (e)): *Point-to-point*, $r = 0.001$, $r = 0.01$, $r = 0.1$, $r = 1$). Node colours depict the different subgroups detected by Louvain community-detection method (electronic supplementary material, Text). Different communities and the overlap between them are also shown by circles around each community. Edge widths depict the edge weights, with thicker edges representing stronger bonds, and finer edges representing weaker bonds. Distance between nodes also depict the strength of connections. Networks were made using spring layout from *NetworkX* package in Python. (a) Point-to-point foraging, (b) central-place foraging; $r = 0.001$, (c) central-place foraging; $r = 0.01$, (d) central-place foraging; $r = 0.1$ and (e) central-place foraging; $r = 1$.

In environments where the habitat quality was lower and patches were more homogeneous in their resource content, foraging units coincided on patches less frequently and for a shorter amount of time, which resulted in fewer interactions (electronic supplementary material, figures S6, S12–S13). At $\beta = 3.5$, when fewer patches were rich, foraging mobility increased with many shorter moves within agents' foraging radii for all radii, while residential mobility increased with longer moves ($\overline{n}_r = 2$, $\overline{d}_r = 0.56$) for $r = 0.1$, but decreased for smaller radii with shorter and similar number of moves ($r = 0.01 : \overline{n}_r = 15$, $\overline{d}_r = 0.041$), or shorter and more frequent moves ($r = 0.001 : \overline{n}_r = 70$, $\overline{d}_r = 0.006$). This effect led to a decrease in both global and local efficiencies across radii ($\overline{E}_{\text{global}} = 0.04$, $\overline{E}_{\text{local}} = 0.3$) from $\beta = 2.5$. However, for radius $r = 0.01$, the decrease in the local efficiencies was lesser when compared with the other radii. For $r = 0.01$, foraging units moved within a space that was small

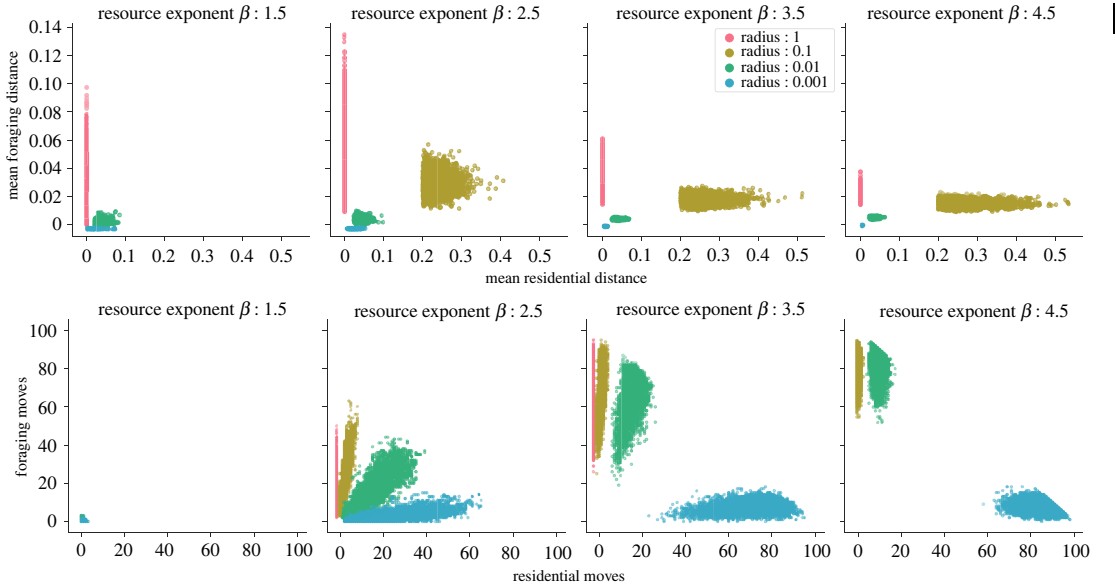

**Figure 4.** Mobility regimes across different environment and radii. Top: the plot shows the mean distance moved in residential moves ($d_r$) against foraging moves ($d_f$). Bottom: the plot shows the frequency of residential moves ($n_r$) and the frequency of foraging moves ($n_f$).

enough to increase the rate of interactions but large enough to find rich patches. When the radius increased ($r = 0.1, 1$) or decreased ($r = 0.001$), foraging units either travelled longer distances and were dispersed in a larger area or were too restricted in their space use to find enough food and continually changed their residence.

Taken together, these data illustrate that CPF units were restricted in their movements, which in turn led to strongly connected subgroups. However, the longer residential moves allowed connections to form between such subgroups to varying extents, whilst this was not possible in the point-to-point model. We found that intermediate levels of overall mobility ($\overline{n} = 23$, $\overline{d} = 0.04$) with few long moves and many shorter moves, for example in $\beta = 2.5$ and $r = 0.1$, created networks that were efficient at both global and local scales. As the frequency of overall movement decreased with longer moves ($r = 1$, $\beta = 2.5: \overline{n} = 15$, $\overline{d} = 0.05$) or increased with shorter moves ($r = 0.01$, $\beta = 2.5: \overline{n} = 34$, $\overline{d} = 0.02$; $r = 0.001$, $\beta = 2.5: \overline{n} = 28$, $\overline{d} = 0.02$), networks lacked dense, long-range connections necessary for global efficiency with either highly locally efficient but fragmented networks or sparsely connected subgroups. Finally, when rate of mobility was very low (highly frequent but very short moves, or rare and short moves), for example in $\beta = 1.5$ and $\beta = 4.5$ (all radii), foraging units rarely interacted with each other, and both global and local efficiencies tended to 0. Altogether, based on our results, we show that CPF with an intermediate radius/mobility regime (between 0.01 and 0.1) maximizes both efficiencies (figure 2$b$). Furthermore, our sensitivity analyses indicate that this result remains robust over longer time-scales (see electronic supplementary material, Results).

## 3.3. Population size and mobility affect network efficiency

We also tested the effect of varying population sizes on the resultant networks by simulating populations of 50, 100 and 200 foraging units. We found that as population size increased, regardless of the radius, the local efficiencies of the networks also increased (figure 5$a$). An increase in population size led to a higher rate of coincidence between foraging units on patches that created denser connections. This effect was stronger when the radius was smaller ($r \leq 0.01$) because the agents were restricted within smaller areas, which led to repetitive interactions and added weight to local connections. Moreover, when the environment was more abundant and heterogeneous ($\beta = 2.5$), agents could spend longer times on rich patches and formed more locally efficient networks.

On the other hand, we found that the effect of population size on global efficiency was not as straightforward (figure 5$b$). In environments with $\beta = 2.5$, global efficiency increased with an increase in population size. This increase was more exaggerated for CPF ($r > 0.001$) and most substantial for intermediate radius ($r = 0.1$). The longer residential moves enabled more foraging units to interact and

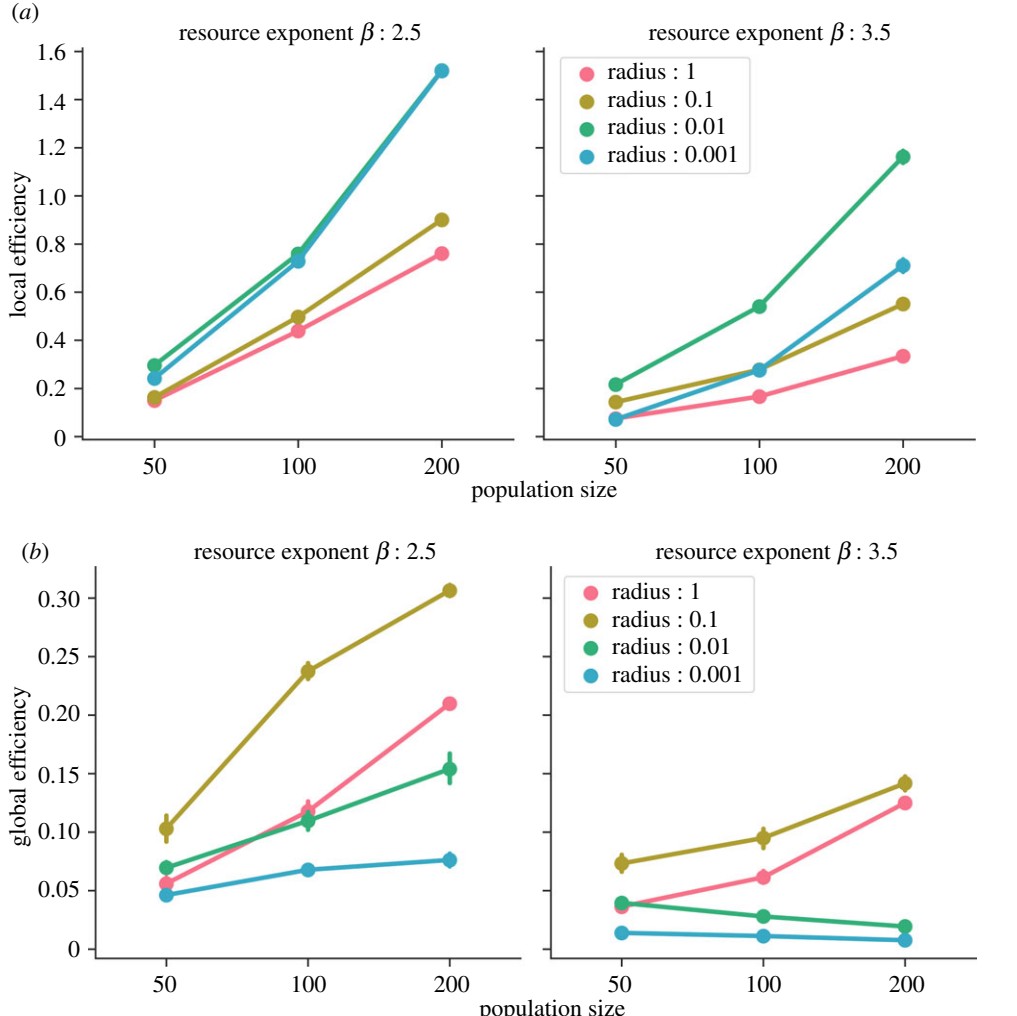

**Figure 5.** Network efficiencies as a function of population sizes after 100 time-steps for $\beta = 2.5$ and $\beta = 3.5$. Error bars indicate 95% confidence intervals. (*a*) Local efficiency and (*b*) global efficiency.

helped create a more connected network. It is also important to note that the networks generated by populations of a small size ($n = 50$) and intermediate foraging radii were more efficient and connected than networks from large population sizes ($n = 200$) that engaged in 'point-to-point' (or $r \leq 0.001$) foraging.

However, for less abundant and more homogeneous environments ($\beta = 3.5$), the effect of population size was diminished and only in the larger radii ($r \geq 0.1$) did it lead to an increase in global efficiencies. When the radii were small or foraging was similar to 'point-to-point', the foraging units experienced lower encounter rates due to shorter residential moves. Without an increase in long-range connections which would have decreased the path length between network nodes, an increase in population size (or number of network nodes) decreased the global efficiency (equation (2.1)) of the networks. When foraging units moved longer distances and foraged within larger radii, long-range connections compensated for a larger network and maintained global connectivity even as population sizes increased. Overall, our results suggest an important role of mobility strategies in mediating the effect of population sizes on information transmission.

## 3.4. Central-place foraging networks are efficient at information transmission

To directly test the networks for their capability of transmitting information, we conducted both simple and complex contagion simulations on the most globally efficient networks that resulted from each model and parameter combination [45].

In line with efficiency results (see previous section), we found that CPF strategies characterized by a combination of residential and foraging moves (e.g. $r = 0.1$ and $\beta = 2.5$) formed networks that allowed a

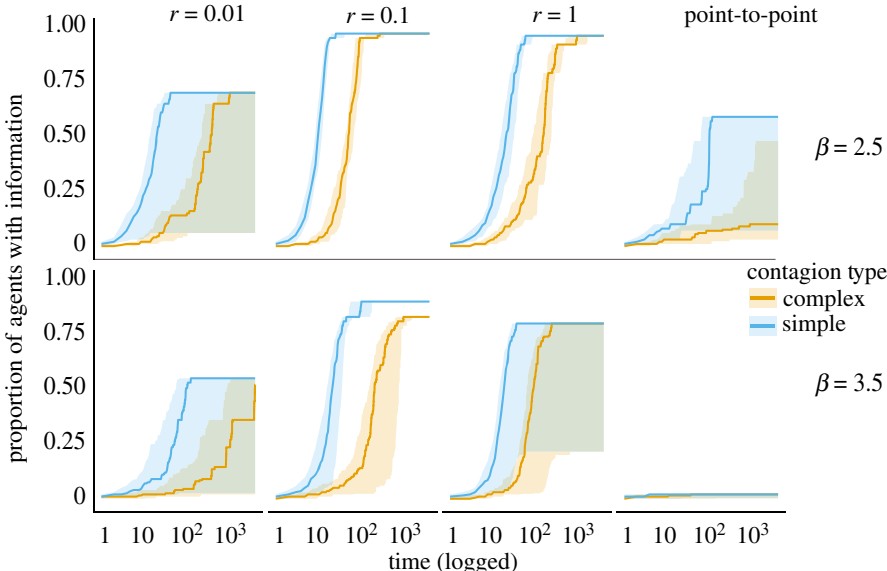

**Figure 6.** Simple and complex contagion trajectories. The plot shows the spread and speed of contagion over time for different radii (columns) and beta (rows). Shaded regions show the 25th and 75th percentiles of the distribution of trajectories at each time-step.

rapid diffusion of information, reaching almost every node. We found that information spread more readily in networks with more extensive and well-connected subgroups when compared with sparser or fragmented networks. For instance, in the point-to-point model, information reached a maximum of around 50% of the population across environments.

In complex contagion, where multiple novel interactions were required for successful transmission of information, we observed a greater effect of network structures and a slower rate of transmission across networks (figure 6). For example, for $r = 0.1$ and $\beta = 2.5$, simple contagion tended to reach 75% of the nodes much faster and more reliably ($\bar{t} = 16 \pm 7$ (s.d.)) than complex contagion which took longer time to reach similar proportions ($\bar{t} = 81 \pm 67$). This effect was magnified for less efficient networks (for example, $r = 1$) where the transmission was much slower and more variable ($\bar{t} = 207 \pm 299$) than in simple contagion ($\bar{t} = 33 \pm 16$) to reach the same proportion (75%) of nodes. In summary, we found that the networks that have high global and local efficiencies (such as those emerging from $\beta = 2.5$ and $r = 0.1$) can maximize both the reach and speed of contagions that resemble cultural transmission.

# 4. Discussion

Recent work on prehistoric and contemporary hunter–gatherer societies has shown that their social networks are efficient at information transmission and could have propelled cultural evolution [5,6,46]. However, the different factors that could have affected the formation of efficient social connectivity are not well understood. In this paper, we assessed how hunter–gatherer foraging patterns could have played a role in the emergence of such efficient social networks. We modelled spatial patterns and mobility regimes emerging from CPF, a derived feature in our lineage, under different environments and tested their implications on the emergence of social networks that are efficient at information transmission [47].

CPF is characterized by foragers bringing back food to central places (homes) while periodically changing the location of such homes according to the availability of resources. Our results reveal that this foraging pattern under most mobility and environmental conditions could have created social networks that are particularly well suited for information exchange. Previous works have suggested that a change in spatial and residence patterns could have caused unique expansions in early hominin social networks [1,46]. We show that, compared with point-to-point foraging, CPF could have modified spatial and residential patterns in ways that would have increased our ancestors' social interactions, made their networks more expansive and improved their ability to exchange information [22]. The main finding from the model by Ramos-Fernández *et al.* [34] showed that interactions between 'point-to-point' foragers following a basic resource-maximization rule could result in structured networks with fission–fusion dynamics. Moreover, previous works have hypothesized that fission–fusion in non-

human primates could have been a precursor for multi-level human social networks [14,24,48,49]. Our results show that the addition of CPF can result in a more extensive fission–fusion, larger and more efficient networks, and suggest one possible pathway that could have partly driven such a transition.

We also find support for the idea that environment-driven variability between the mobility regimes employed by different hunter–gatherer societies has significant consequences for their social networks and hence cultural transmission [10,18,50,51]. Similar to Perreault & Brantingham [35], we find that mobility regimes which combine short-scale foraging and long-scale residential movements can create more efficient networks as opposed to regimes that are primarily residential or sedentary. In heterogeneous environments, when central-place foragers' movements are restricted within an intermediate radius with occasional long residential moves to richer resource patches, the networks formed contain densely connected subgroups embedded in more extensive regional networks. Our results predict that an intermediate mobility regime (figure 2b), thus, could balance the trade-off between networks that are highly cliquish at the expense of global efficiency and sparser large networks with low clustering. Such networks, similar to small-world topologies, can support information processing at local and global scales [4,52,53].

In line with previous research highlighting the importance of demography for cultural evolution, we show that, under most circumstances, an increase in population density can result in more efficient networks and a larger capacity for information exchange [54–56]. At the same time, we find support for previous predictions that mobility, in addition to population density, plays an important role in affecting cultural transmission [56,57]. We show that residential mobility and CPF can improve connectivity even in small populations [58], and can generate networks that are as efficient as the networks from large populations engaged in 'point-to-point foraging'. Hunter–gatherer groups with low population densities could have therefore increased their mobility to maintain encounter rates that would have kept them viable by allowing better connectivity, promoting exogamy, efficient exchange of information and resilience to climatic variation [46,59]. Thus, our results also emphasize the importance of optimal connectivity and mobility within a population to offset the adverse effects of demographic collapses on cultural transmission [33,54].

Our work reveals that ecologically driven foraging and mobility decisions can generate networks that resemble the structure and composition of networks observed in real hunter–gatherer societies. The agents or foraging units in our model represented nuclear families which across hunter–gatherer societies normally comprise around 4–5 individuals [36,60]. We found that in environments with intermediate heterogeneity ($\beta = 2.5$), CPF with intermediate radii ($r = 0.1$) which afforded local interactions within overlapping foraging radii and global interactions due to longer residential moves formed networks with multiple and nested levels. More specifically, the emergent networks fused foraging units into different (approx. 15) subgroups (analogous to bands of co-residing family units) that were composed of 5–7 foraging units each, and on an average half of these subgroups (7–9) were interconnected with sparse ties forming a higher level of organization of approximately 40 foraging units (analogous to communities or mega-bands). Such network organization is similar to ethnographic reports across 336 contemporary hunter–gatherer societies [36,60] and estimations based on energetic constraints [5,13,18,31,37] that show hunter–gatherer regional metapopulations of 100 families that can be fragmented into co-residing bands of approximately 10 families, which are in turn interconnected and form a larger community (approx. 3–4 co-residing bands or approx. 30 families) within the metapopulation. For better comparisons with empirical data on social organization, future studies can base their models on empirical ecological or mobility data and investigate the emergence of multi-level sociality in more detail.

These findings hold significant implications for our species' evolutionary history and the ability to develop cumulative culture [61]. The degree and strength of intra- and inter-regional group interactions among prehistoric hunter–gatherers and their spatial distribution have been proposed to be key factors for cultural transmission [26]. The focus of our model was on social network patterns that can arise solely from the derived features of hunter–gatherer foraging-related mobility in different environments, as we wished to unravel their implications for information transmission. Accordingly, we did not consider other social factors that could have shaped their mobility decisions (such as cooperative breeding, resource sharing or joint ritual participation), further structured their interaction networks or potentially resulted in greater incentives and/or efficiency of information transmission. Nonetheless, the model sheds light on the mechanisms by which the regional-scale connectivity generated by individual CPF despite low population sizes throughout our species history. Such connectivity could have maintained cultural diversity and complexity by allowing cultural recombination, transmission of innovations and preventing the loss of existing culture [2,5,11,62,63].

Further research could elaborate on more complex portrayals of physical (for example, resource distribution, travelling costs, seasonality) and social (for example, demography, inter-forager competition, cooperation, sociality, kinship, learning) environments that would have characterized early hunter–gatherer communities. These factors would have potentially interacted with foraging and mobility decisions and cultural complexity [28,64–67]. Moreover, these factors would have also interacted with the cognitive capacities of our early ancestors (e.g. spatial memory, longer-range planning, larger neocortex, theory of mind, symbolic communication) [68,69]. Such cognitive factors would have affected the ability to explore larger spaces, engage in CPF and maintain more extensive social networks, and possibly created selection pressures that paved the way for present-day human cognition and culture [70–74].

Although additional studies should also address potential selection pressures experienced by our ancestors that would have led them to start using and returning to central places, our study corroborates early claims that CPF would have had important implications for the accumulation and transmission of tools and other types of information [12]. However, our work highlights the role of mobility and spatial patterns that stem from CPF in our evolutionary history. We suggest that mobility-driven networks could have led to positive feedback whereby a more efficient transmission of social and/or ecological information, increased food sharing, better resource defences, and a greater accumulation of material culture at a few places would have been advantageous to central-place foragers [39,75]. This advantage could have further promoted reliance on increasingly complex culture and encouraged adaptations to social networks (for example, through kinship or trade) to efficiently generate, transmit and sustain such culture [11,37,58,76–78].

Data accessibility. The agent-based model was run in Python 3. Contagion simulations were done using R, and other analyses were done in both Python and R. ODD description of the model, and relevant data and code for this research work are stored in GitHub: https://github.com/ketikagarg/information_transmission and have been archived within the Zenodo repository: https://doi.org/10.5281/zenodo.5655317 [79].

Competing interests. We declare we have no competing interests.

Funding. This work was been partially supported by the Diverse Intelligences Summer Institute, whose programs are funded by TWCF Grant 0333 to UCLA; the 'Fundación La Caixa' research fellowship (to C.P.-I) and University of Zurich Forschungskredit CANDOC grant FK-19-083 award (to C.P.-I).

Acknowledgements. We thank the Diverse Intelligences Summer Institute (DISI 2020) for inspiring this project, for giving us an opportunity to work together and for funding our work. We also thank Dieter Lukas and anonymous reviewers and editors, Robert Foley, Jacob Foster, Luke Premo and Paul Smaldino and his lab for their helpful feedback and comments. We are also grateful to Luisa Espinós for help with model visualization and Andrea Migliano for support.

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
