## [Peer Review File · Royal Society Open Science]

Review History

RSOS-211324.R0 (Original submission)

Review form: Reviewer 1

Is the manuscript scientifically sound in its present form?

Yes

Are the interpretations and conclusions justified by the results?

Yes

Is the language acceptable?

Yes

Do you have any ethical concerns with this paper?

No

Have you any concerns about statistical analyses in this paper?

No

Recommendation?

Accept as is

Comments to the Author(s)

We think the authors have responded well to our initial concerns (as Reviewer 3), and we thank them for their detailed responses to our comments. The responses to our original point 11 and our point concerning the ODD protocol are particularly welcome and we think they strengthen the paper. We see no reason not to publish the paper as it currently stands.

Review form: Reviewer 2

Is the manuscript scientifically sound in its present form?

Yes

Are the interpretations and conclusions justified by the results?

Yes

Is the language acceptable?

Yes

Do you have any ethical concerns with this paper?

No

Have you any concerns about statistical analyses in this paper?

No

Recommendation?

Accept with minor revision (please list in comments)

Comments to the Author(s)

The topic of the manuscript, as well as the methods chosen are interesting and will provide a useful contribution to the literature on the topic. However, I believe that some important information is still missing in the manuscript's current form. I understand that this manuscript has already gone through two rounds of reviews, and I can see that the additions done after each have strengthened the paper. However, as a first time reader of this paper, I still have some questions that I think should be addressed.

In general, the methodology seems sound, the conclusions are supported by the data, and the results are interesting. Here are the points that should be improved:

1. I believe that more information is needed about the model(s) used for this research:

1.1. In the description of the model, the authors do not clarify what happens if two foraging-units have overlapping radii or if they choose the same cell as their homebase. From the response to reviewers, I was able to see that such situations are not avoided. When I think about it, it makes sense that overlapping radii would be good as this is when interactions occur, but this should be mentioned in the methods section.

1.2. How are resources used in the model? On line 75, the authors say that the landscape is a "two-dimensional environment that ranges from 0 to 1." What does the range represent? I assume it is resources, but as line 84 says that resources are depleted by "one unit" at every time step, I am not sure about it. What does that unit represent in line 84?

1.3. How did the authors run the point-to-point model? This is not explained in the methods section. Did they run the model that Ramos-Fernández wrote or did they use specific settings in their own ABM to mimic Ramos-Fernández's model?

1.4. While the code is available (which is great), it would be good to indicate what language the model is written in within the text. This also applies to the analysis (was it Python or R for the different tests).

1.5. The section describing the model has sentences that use the past tense and others that are in the present. It would make for easier reading if all verbs were in the same tense.

1.6. Line 81 has the sentence "k has a broad range with high values of k" which is confusing. I would rephrase for something like "k has a broad range of high values."

2. I think that the way the transmission tests were done should be explained a bit more.

2.1. In particular, was this done within the ABM or after, on its outputs? If the latter, which language (R, Python) or program was used to do so? Moreover, how was the test set up? Was one node "infected" at random and then time went until 5000 time-steps? If, did you repeat the tests multiple times to see how different "starting infections" affected the results? This section could use more details.

2.2. On line 141-143, they explain that for "simple transmission, a single interaction event is sufficient for transmission," but then they say that the transmission is based on the number of people with the information and the strength of the connection. To me, these two things seem to contradict each other (if a single interaction is sufficient, the strength of the connection should not matter). So, this should be rephrased to clarify.

3. The model has one parameter that controls two environmental settings (richness and heterogeneity), therefore the authors only look at two scenarios (high-high and low-low), but why did they not try the other two combinations (high-low, low-high)? Given the high impact of environment on networks, those two additional scenarios might provide different results. Here, I am not saying that the authors should rerun the model, but rather that they should at least discuss why they did not consider those scenarios.

4. The discussion has a few statements that need to be nuanced or better explained:

4.1. Line 276, the authors say that "the results reveal that central-place foraging could have created social networks that are particularly suited for information exchange." First, I think the word "well" is missing, but I may be wrong. Second and more important here, the results show this, but only when the environment is at intermediate heterogeneity and when mobility is intermediate as well, so central-place foraging does not always create social networks that lead to good transmission. This statement should be nuanced a bit.

4.2. Lines 296-297 is similar. The authors say that population increase will lead to more transmission. However, the results show that it is more nuanced than that, as global efficiency does not correlate well with population size.

4.3. Lines 298-299, the authors say that population size and mobility have an equally important role in the creation of social networks, but it is unclear which results support that, as the contribution of different parameters were not directly compared.

And a few additional miscellaneous details:

5.1. Line 19: "One of the pivotal transitions in human evolution is our ability to innovate..." The ability itself is not a transition. The innovation of this ability is. So, this should be rephrased.

5.2. Line 31: "shorter foraging trips" shorter than who?

5.3. Figure 2: Why use an inset for the relationship between efficiencies and radius? At the moment, its placement makes it look like this graph focuses on local efficiency, but that is not the case. Also, the labels are very small in that inset, so I would revise the layout of this figure.

5.4. Figure 3: The choice of colors for the groups is too reduced. It makes it look like multiple groups share something that is not explained here (as there is not enough difference between

some of the colors). I would expand the color scale to have more variability (I know it's tricky when trying to keep things colorblind friendly, though).

5.5. Figure 3 caption: It says it gives the graphs' legend clockwise, but that's not really clockwise, so I would use a different term.

5.6. Figure 4: Here, it is hard to see the data in the left plots. Maybe adding transparency would show that the data is all there, but overlapping.

5.7. Lines 216-217: in the parentheses, why the two different sets of values? What do they represent?

This research is very interesting and I think that the results are significant. I just feel it still needs some work to be more easy to understand.

Decision letter (RSOS-211324.R0)

Dear Ms Padilla-Iglesias

On behalf of the Editors, we are pleased to inform you that your Manuscript RSOS-211324 "Hunter-gatherer foraging networks promote information transmission" has been accepted for publication in Royal Society Open Science subject to minor revision in accordance with the referees' reports. Please find the referees' comments along with any feedback from the Editors below my signature.

Please submit your revised manuscript and required files (see below) no later than 7 days from today's (ie 28-Oct-2021) date. Note: the ScholarOne system will 'lock' if submission of the revision is attempted 7 or more days after the deadline. If you do not think you will be able to meet this deadline please contact the editorial office immediately.

on behalf of Dr Dieter Lukas (Associate Editor) and Kevin Padian (Subject Editor)
openscience@royalsociety.org

Associate Editor Comments to Author (Dr Dieter Lukas):

Comments to the Author:

Dear authors,

Your article entitled "Hunter-gatherer foraging networks promote information transmission" has now been seen by two reviewers and the reviewers' comments are appended below. Thank you for making the edits to your manuscript in response to my (and the previous reviewers' comments). I agree with reviewer 1 that you have done a great job in addressing the issues that were previously raised. Reviewer 2 highlights a few more points asking for some further details and clarifications. This is again in the spirit of helping the reader to follow all the decisions you made and to understand the transferability of the results across systems and approaches. As mentioned before, RSOS does not have any restrictions on manuscript length/format. While the provided code makes it clear what you did (thank you for making this open), it would be helpful if you could also provide these details in the manuscript.

Even though this is labelled a "major revision" I am expecting that it should be relatively easy for you to make all necessary changes. It just means that I want to have another quick look at it, and for that the only option in the system is to choose "major revisions". While I trust that you will make the necessary edits, I think it might be helpful to have another check in case there were any misunderstandings. I am looking forward to the edited manuscript.

Reviewer comments to Author:

Reviewer: 1

Comments to the Author(s)

We think the authors have responded well to our initial concerns (as Reviewer 3), and we thank them for their detailed responses to our comments. The responses to our original point 11 and our point concerning the ODD protocol are particularly welcome and we think they strengthen the paper. We see no reason not to publish the paper as it currently stands.

Reviewer: 2

Comments to the Author(s)

The topic of the manuscript, as well as the methods chosen are interesting and will provide a useful contribution to the literature on the topic. However, I believe that some important information is still missing in the manuscript's current form. I understand that this manuscript has already gone through two rounds of reviews, and I can see that the additions done after each have strengthened the paper. However, as a first time reader of this paper, I still have some questions that I think should be addressed.

In general, the methodology seems sound, the conclusions are supported by the data, and the results are interesting. Here are the points that should be improved:

1. I believe that more information is needed about the model(s) used for this research:

1.1. In the description of the model, the authors do not clarify what happens if two foraging-units have overlapping radii or if they choose the same cell as their homebase. From the response to reviewers, I was able to see that such situations are not avoided. When I think about it, it makes sense that overlapping radii would be good as this is when interactions occur, but this should be mentioned in the methods section.

1.2. How are resources used in the model? On line 75, the authors say that the landscape is a "two-dimensional environment that ranges from 0 to 1." What does the range represent? I assume

it is resources, but as line 84 says that resources are depleted by "one unit" at every time step, I am not sure about it. What does that unit represent in line 84?

1.3. How did the authors run the point-to-point model? This is not explained in the methods section. Did they run the model that Ramos-Fernández wrote or did they use specific settings in their own ABM to mimic Ramos-Fernández's model?

1.4. While the code is available (which is great), it would be good to indicate what language the model is written in within the text. This also applies to the analysis (was it Python or R for the different tests).

1.5. The section describing the model has sentences that use the past tense and others that are in the present. It would make for easier reading if all verbs were in the same tense.

1.6. Line 81 has the sentence "k has a broad range with high values of k" which is confusing. I would rephrase for something like "k has a broad range of high values."

2. I think that the way the transmission tests were done should be explained a bit more.

2.1. In particular, was this done within the ABM or after, on its outputs? If the latter, which language (R, Python) or program was used to do so? Moreover, how was the test set up? Was one node "infected" at random and then time went until 5000 time-steps? If, did you repeat the tests multiple times to see how different "starting infections" affected the results? This section could use more details.

2.2. On line 141-143, they explain that for "simple transmission, a single interaction event is sufficient for transmission," but then they say that the transmission is based on the number of people with the information and the strength of the connection. To me, these two things seem to contradict each other (if a single interaction is sufficient, the strength of the connection should not matter). So, this should be rephrased to clarify.

3. The model has one parameter that controls two environmental settings (richness and heterogeneity), therefore the authors only look at two scenarios (high-high and low-low), but why did they not try the other two combinations (high-low, low-high)? Given the high impact of environment on networks, those two additional scenarios might provide different results. Here, I am not saying that the authors should rerun the model, but rather that they should at least discuss why they did not consider those scenarios.

4. The discussion has a few statements that need to be nuanced or better explained:

4.1. Line 276, the authors say that "the results reveal that central-place foraging could have created social networks that are particularly suited for information exchange." First, I think the word "well" is missing, but I may be wrong. Second and more important here, the results show this, but only when the environment is at intermediate heterogeneity and when mobility is intermediate as well, so central-place foraging does not always create social networks that lead to good transmission. This statement should be nuanced a bit.

4.2. Lines 296-297 is similar. The authors say that population increase will lead to more transmission. However, the results show that it is more nuanced than that, as global efficiency does not correlate well with population size.

4.3. Lines 298-299, the authors say that population size and mobility have an equally important role in the creation of social networks, but it is unclear which results support that, as the contribution of different parameters were not directly compared.

And a few additional miscellaneous details:

5.1. Line 19: "One of the pivotal transitions in human evolution is our ability to innovate..." The ability itself is not a transition. The innovation of this ability is. So, this should be rephrased.

5.2. Line 31: "shorter foraging trips" shorter than who?

5.3. Figure 2: Why use an inset for the relationship between efficiencies and radius? At the moment, its placement makes it look like this graph focuses on local efficiency, but that is not the case. Also, the labels are very small in that inset, so I would revise the layout of this figure.

5.4. Figure 3: The choice of colors for the groups is too reduced. It makes it look like multiple groups share something that is not explained here (as there is not enough difference between some of the colors). I would expand the color scale to have more variability (I know it's tricky when trying to keep things colorblind friendly, though).

5.5. Figure 3 caption: It says it gives the graphs' legend clockwise, but that's not really clockwise, so I would use a different term.

5.6. Figure 4: Here, it is hard to see the data in the left plots. Maybe adding transparency would show that the data is all there, but overlapping.

5.7. Lines 216-217: in the parentheses, why the two different sets of values? What do they represent?

This research is very interesting and I think that the results are significant. I just feel it still needs some work to be more easy to understand.

===PREPARING YOUR MANUSCRIPT===

one version should clearly identify all the changes that have been made (for instance, in coloured highlight, in bold text, or tracked changes);

===PREPARING YOUR REVISION IN SCHOLARONE===

-- Ensure that your data access statement meets the requirements at <https://royalsociety.org/journals/authors/author-guidelines/#data>.

You should ensure that you cite the dataset in your reference list. If you have deposited data etc in the Dryad repository, please only include the 'For publication' link at this stage. You should remove the 'For review' link.

-- If you are requesting an article processing charge waiver, you must select the relevant waiver option (if requesting a discretionary waiver, the form should have been uploaded, see 'File upload' above).

-- If you have uploaded any electronic supplementary (ESM) files, please ensure you follow the guidance at <https://royalsociety.org/journals/authors/author-guidelines/#supplementary-material> to include a suitable title and informative caption. An example of appropriate titling and captioning may be found at https://figshare.com/articles/Table_S2_from_Is_there_a_trade-off_between_peak_performance_and_performance_breadth_across_temperatures_for_aerobic_sc_ope_in_teleost_fishes_/3843624.

Author's Response to Decision Letter for (RSOS-211324.R0)

See Appendix A.

Decision letter (RSOS-211324.R1)

Dear Ms Padilla-Iglesias,

It is a pleasure to accept your manuscript entitled "Hunter-gatherer foraging networks promote information transmission" in its current form for publication in Royal Society Open Science. The comments of the reviewer(s) who reviewed your manuscript are included at the foot of this letter.

on behalf of Dr Dieter Lukas (Associate Editor) and Kevin Padian (Subject Editor)
openscience@royalsociety.org

Associate Editor Comments to Author (Dr Dieter Lukas):

Thank you for addressing all the comments of the reviewer. The manuscript now provides the relevant methodological details and careful framing that helps readers understand the important contribution this research is making.

Appendix A

Dear Dr Dieter Lukas,

Thank you very much for your interest in our manuscript and for all of your feedback. We have now tackled the points raised by reviewer 2, mainly increasing the clarity with which the model is described and its outputs analysed. We have also modified the figures according to the Reviewer's suggestions. As before, the responses to the specific comments by the reviewers are detailed below. We have also highlighted the new modifications on the main manuscript document.

Reviewer: 2

Comments to the Author(s)

The topic of the manuscript, as well as the methods chosen are interesting and will provide a useful contribution to the literature on the topic. However, I believe that some important information is still missing in the manuscript's current form. I understand that this manuscript has already gone through two rounds of reviews, and I can see that the additions done after each have strengthened the paper. However, as a first time reader of this paper, I still have some questions that I think should be addressed.

In general, the methodology seems sound, the conclusions are supported by the data, and the results are interesting. Here are the points that should be improved:

1. I believe that more information is needed about the model(s) used for this research:

1.1. In the description of the model, the authors do not clarify what happens if two foraging-units have overlapping radii or if they choose the same cell as their homebase. From the response to reviewers, I was able to see that such situations are not avoided. When I think about it, it makes sense that overlapping radii would be good as this is when interactions occur, but this should be mentioned in the methods section.

Indeed, we agree with the reviewer that not only specifying the "possibility" of radii overlap is relevant but also its relationship with the literature (and hence our motivation behind coding the model the way we did). To do so, we have now included a paragraph clarifying this under the model description section. Here, as we did in the response to the editor, we explain that each time-step that a foraging-unit coincided with another foraging-unit on a patch, they formed a social network tie or added a unit of weight to an existing tie. We also expand on pre-existing literature indicating the prevalence of family units sharing home patches and foraging patches.

1.2. How are resources used in the model? On line 75, the authors say that the landscape is a "two-dimensional environment that ranges from 0 to 1." What does the range represent? I

assume it is resources, but as line 84 says that resources are depleted by "one unit" at every time step, I am not sure about it. What does that unit represent in line 84?

We now clarify this by mentioning that patches are depleted by a unit in their resource content (k) at every time step, and clarified that 0-1 concerns the spatial position of each patch, and is not related to its resource content.

1.3. How did the authors run the point-to-point model? This is not explained in the methods section. Did they run the model that Ramos-Fernández wrote or did they use specific settings in their own ABM to mimic Ramos-Fernández's model?

We have now clarified that when the radius was 0 in our model, the parameters were equivalent to those from Ramos-Fernández's model. Nonetheless, we also ran their original model to corroborate our results.

1.4. While the code is available (which is great), it would be good to indicate what language the model is written in within the text. This also applies to the analysis (was it Python or R for the different tests).

We have now added this in the Data and Code Availability section.

1.5. The section describing the model has sentences that use the past tense and others that are in the present. It would make for easier reading if all verbs were in the same tense.

Thanks a lot for this suggestion. We have now fixed this so that it is all in the past tense.

1.6. Line 81 has the sentence "k has a broad range with high values of k" which is confusing. I would rephrase for something like "k has a broad range of high values."

We have now amended this accordingly.

2. I think that the way the transmission tests were done should be explained a bit more.

2.1. In particular, was this done within the ABM or after, on its outputs? If the latter, which language (R, Python) or program was used to do so? Moreover, how was the test set up? Was one node "infected" at random and then time went until 5000 time-steps? If, did you repeat the tests multiple times to see how different "starting infections" affected the results? This section could use more details.

We thank the reviewer for pointing this out. We have now mentioned these important details in the Methods section.

2.2. On line 141-143, they explain that for "simple transmission, a single interaction event is sufficient for transmission," but then they say that the transmission is based on the number of people with the information and the strength of the connection. To me, these two things seem to contradict each other (if a single interaction is sufficient, the strength of the connection should not matter). So, this should be rephrased to clarify.

Thank you very much for raising this issue. We have now re-written the text on complex contagion on page 4 as well as included examples in order to clarify the differences between simple and complex contagion models, and the role of edge weights on each. We clarify that the probability of transmission in both models depended upon the edge weights and the number of neighbors with the information/behavior to be transmitted. Compared to simple contagion, complex contagion required stronger ties and/or more exposure to neighbors with information for successful adoption.

3. The model has one parameter that controls two environmental settings (richness and heterogeneity), therefore the authors only look at two scenarios (high-high and low-low), but why did they not try the other two combinations (high-low, low-high)? Given the high impact of environment on networks, those two additional scenarios might provide different results. Here, I am not saying that the authors should rerun the model, but rather that they should at least discuss why they did not consider those scenarios.

Since we wanted to compare our model with that of Ramos-Fernández and colleagues, we only tested the environmental configurations used by Ramos-Fernández and colleagues, as an "experimental control". Our aim was to primarily test central-place foraging behavior on social networks, and to shed light on the fact that environmental features need to be taken into consideration when evaluating such effects. However, we agree that potentially different environmental configurations would yield different results, and therefore future research aiming at specifically pinpointing the precise effects of particular environmental settings on social networks (for example, to try to reconstruct ancient social networks in different parts of the world at different points in time) should take those into consideration. We briefly discuss this in the penultimate paragraph of the discussion section.

4. The discussion has a few statements that need to be nuanced or better explained:

4.1. Line 276, the authors say that "the results reveal that central-place foraging could have created social networks that are particularly suited for information exchange." First, I think the word "well" is missing, but I may be wrong. Second and more important here, the results show this, but only when the environment is at intermediate heterogeneity and when mobility is intermediate as well, so central-place foraging does not always create social networks that lead to good transmission. This statement should be nuanced a bit.

We apologise for the over-generalisation, as indeed population density increases the efficiency of information transmission under most mobility and environmental conditions but not all. Consequently, we have changed this statement to: ‘Our results reveal that this foraging pattern under most mobility and environmental conditions could have created social networks that are particularly well-suited for information exchange.’

4.2. Lines 296-297 is similar. The authors say that population increase will lead to more transmission. However, the results show that it is more nuanced than that, as global efficiency does not correlate well with population size.

We have added a clause here to not generalize - Similar to previous research highlighting the importance of demography for cultural evolution, we find that, under most conditions, an increase in population density can result in more efficient networks and a larger capacity for information exchange.

4.3. Lines 298-299, the authors say that population size and mobility have an equally important role in the creation of social networks, but it is unclear which results support that, as the contribution of different parameters were not directly compared.

We have removed the word ‘equally’ - Our results support previous predictions that population density is not the sole explanation for cultural transmission, and mobility plays an important role.

And a few additional miscellaneous details:

5.1. Line 19: “One of the pivotal transitions in human evolution is our ability to innovate...” The ability itself is not a transition. The innovation of this ability is. So, this should be rephrased.

We believe this statement has been misinterpreted. What we meant here is that a pivotal transition in human evolution is indeed the capacity of humans to come up with (or innovate) complex cultural traits, accumulate them over time and be dependent on them. For clarity, we have now changed the word “innovate” for “generate” as it makes direct reference to the cultural traits themselves.

5.2. Line 31: “shorter foraging trips” shorter than who?

We have now changed “shorter” for “short”.

5.3. Figure 2: Why use an inset for the relationship between efficiencies and radius? At the moment, its placement makes it look like this graph focuses on local efficiency, but that is not the case. Also, the labels are very small in that inset, so I would revise the layout of this figure.

Thank you, we have now split this figure into an equal-sized, 3-way panel plot.

5.4. Figure 3: The choice of colors for the groups is too reduced. It makes it look like multiple groups share something that is not explained here (as there is not enough difference between some of the colors). I would expand the color scale to have more variability (I know it's tricky when trying to keep things colorblind friendly, though).

We have expanded the color scale more now.

5.5. Figure 3 caption: It says it gives the graphs' legend clockwise, but that's not really clockwise, so I would use a different term.

Thank you for this. We have removed this now.

5.6. Figure 4: Here, it is hard to see the data in the left plots. Maybe adding transparency would show that the data is all there, but overlapping.

We thank the reviewer for this suggestion, but the current plot is with minimum transparency but due to a large number of data points, it appears this way. Therefore, respectfully, we have decided to leave the plot as it is, as we believe that the presentation shows the contrast in the distribution of points across different conditions, especially around the most significant results, like $B=2.5, 3.5$

5.7. Lines 216-217: in the parentheses, why the two different sets of values? What do they represent?

We are discussing the importance of $r=0.1$ mobility and comparing it to other conditions. First we mention the $r=1$ condition which decreases the frequency of movement and increases longer moves. Then, we mention that, on the other hand, smaller radii, $r=0.01, 0.001$, increases the frequency of movement with shorter moves.

Please let us know if there is anything else you would like us to revise. We look very much forward to your decision.

Sincerely,

Cecilia Padilla-Iglesias, Ketika Garg, Nicolás Restrepo Ochoa and Bleu Knight

Dear Editor,

First of all, thank you for your detailed feedback and suggestions, as well as for your interest in our manuscript. We believe now, after addressing the points you and the previous reviewers suggested, the paper presents a much stronger contribution to understanding the role of foraging behaviours on the formation of social networks and their properties. We have expanded on the methodology as well as included ethnographic justifications for our modeling decisions and for contextualising the model results. The detailed responses of the comments you raised are included below, as well as the responses we gave to the previous reviewers. We have highlighted all of the important changes and additions in the manuscript.

Comments from RSOS editor

You appear to have made two responses to "the fact that individuals live in groups": first, you argue that you want your model to lead to the formation of group in response to foraging decisions; and second, you argue that collective group foraging could be a future addition to your model.

Your argument to address the first issue is that the aim of your model is to see whether the need to solve foraging problem in particular ecologies leads to the formation of groups. However, for all we know, ancestral human societies always lived in groups, and central-place foraging most likely was an adaption to deal with the competition that emerges when foraging in a group-setting. In your reply you talk a lot about derived features. I am not sure what you are basing this argument on that group-living is a recently derived feature of humans. All the evidence I have seen is that group-living existed in humans since before the split from the common ancestor with the African great apes. The group aspect is key, because it leads both to a difference in the social networks (also see my next paragraph) and because it changes the question. It is not whether ecology leads to group formation but whether, given the ecologies they experienced, humans forage in groups like baboons which is central place foraging in groups; or maybe like gorillas, which is roaming in groups with frequent encounters between groups at high-quality foraging patches; or maybe, like what you seem to be focusing on, chimpanzees with random movement within shared community territory. I think your model has the potential to show how these difference foraging strategies can emerge depending on the ecological conditions and what influence they have on the social connections that form. However, for this you have to make that explicit what your baseline model is, what other variation look like, and what the social structure is. Thinking about these different scenarios might also help you to consider more the multilevel structure you repeatedly refer to. Currently, I

am not sure that your model generates a multilevel structure - it rather seems to reflect a single community in which individuals have varying degrees of connectivity (like a single chimpanzee community). Recognizing from the beginning in your model that humans live in groups and partitioning individuals accordingly (see below) might also get at the issue of the difference between connections within a group and links between groups.

With regard to your second response of an addition of a paragraph in the discussion on collective foraging, the distinction between individual and group foraging is not what the two reviewers were focusing on. Their point is that the home, the central place from which foraging starts, is usually shared by multiple individuals that then go out and forage individually. The issue is that the social connections individuals form in foraging societies are likely to happen in a fundamentally different from what your model assumes. The observation that individuals in foraging societies live in groups and return to these groups after foraging trips implies that most connections happen at the home. For example, this recent article shows that Hadza come together with many others in camps, but that during foraging men only have 7% chance to be near another individual (<https://www.nature.com/articles/s41562-020-01002-7.pdf>). Your model sets up the opposite: all social connections are formed during the foraging trips when individuals exploit the same patch, after which individuals return home alone.

It seems to me like there could be different options to address this key issue of human social structure. One would be to change the setup of the model. Currently you state that "The model is initialized with foragers independently and randomly distributed across the patches." I wonder if you could change this to initialise the model in a way that multiple (~15) individuals will start at the same patch. The following behaviour of the individuals could remain unchanged, individuals forage independently from their home base and might decide to move on once they have exploited a patch. In that case a movement though would involve everyone at the patch to keep the group structure intact. This seems like a relatively simple change to add to the model, but one that might lead to fundamentally different social networks. Given that the social networks are the focus of your article, excluding the inherently social structure of human groups will generate outcomes that are difficult to link to the actual behaviour of humans.

We agree that a basic level of social structure would have been present in the earliest hunter-gatherers. To address this issue perhaps it is clearer to reframe our model so that each agent can be conceptualised as a foraging/nuclear family unit (i.e. the basic unit of social organization in hunter-gatherer societies; see Lewis et al. 2014, *Nature Comms* for an example of a similar approach). Nuclear families where a few individuals go out to forage to provision for the rest of the family members represent the basic unit of production and consumption. Hence, any ties formed by any two agents would therefore be considered to be a tie between two family units. Since such core units do not disband, for simplification, we can consider these families as a single, co-residing core unit. At the same time these core units have been well-documented to regularly interact, fuse and disband to form a higher level of organization (bands or camps), which our model simulates (Migliano et al. 2017;2021; Kelly et al. 2013; Binford et al. 1980).

Such bands have been shown to share the same home-bases (or central places) and often co-reside, which leads to a dense network of interactions between them. In our model, when two foraging-units coincide on a home-patch, they share the same home range, forage together, and form strong connections, leading to the creation of sub-groups (and hence giving a modular structure to the network). In addition, if two foraging-units share a foraging-patch (in contrast to a home-patch), they share overlapping home ranges which also have been documented to lead to formation of larger communities (Hamilton et al. 2007;2018). As a result, each family exists in a small-world of dense local interaction (bands), but are connected to the larger network by sparse global interactions which resembles a multi-level structure and could have been a precursor to more established and efficient multi-level sociality (Grueter et al. 2020). Thus, our model simulates how foraging and mobility decisions affect the encounter rates of foraging units (or families) within a metapopulation and thus its social network, and the ability to efficiently transmit information.

We have now added a paragraph in the Discussion discussing the general multi-level structure, and the social organization that emerges in our results (Lines 296-308). More specifically, we argue that the interactions between foraging-units (which for our model we report in Figures S6 and S8) give rise to a “band structure” and determine the number of sub-groups (bands) within the regional community and the average number of individuals in each band. Across hunter-gatherer societies, foraging units or nuclear families, which in our model are represented by agents, normally comprise around 4-5 individuals. In the $\beta=2.5$ environment, this leads to the establishment of subgroups (which can be seen as bands of co-residing families) that range between 5-7 agents (or family units) which is similar to both ethnographic reports of band sizes across 336 hunter-gatherer societies as well as estimated optimal band sizes for hunter-gatherers based on energetic constraints (see Kelly et al. 2013 and Hamilton, 2021; 2007; Marlowe, 2005; Binford 2001). In turn, residential camps tend to be connected on average to 7-9 other residential camps, representing hunter-gatherer communities. This is also similar to the ethnographic findings showing that hunter-gatherer communities on average comprise around 9-12 families (Hamilton et al. 2007; Hamilton, 2021). The metapopulation size of 100 agents that is normally taken as representing the average size of hunter-gatherer regional groups. It is also estimated that around 4 of such regional populations may comprise overarching “metapopulations” (Hamilton et al. 2021; Bird et al. 2019; Hill et al. 2014), but there is a lack of empirical research on both their size and composition.

With regards to the point raised about our baseline model, i.e. the model by Ramos-Fernandez et al, their main finding is the fact that they are able to observe structured networks with fission-fusion dynamics *solely* from interactions between agents following simple foraging-related rules. We explain their results more in detail on the first paragraph of the Methods section (starting on Line 67). The addition of central-place foraging in our model resulted in a more extensive fission-fusion, larger and more efficient networks. Previous works have hypothesized that fission-fusion in other primates could have been a precursor for multi-level human social structures (Layton 2012, Aureli et al. 2008). Thus, our results suggest a

possible mechanism that could have caused that transition. We have also added this to our discussion (Line 275).

Lastly, we also wish to clarify that foraging-units in our model can coincide both at home-patches and other patches. In both scenarios, either due to the same home-range or overlapping home-ranges, respectively, they continue to form connections with each other due to interacting more frequently within shared spaces. We believe that this is a good approximation to a more detailed model which distinguishes between diurnal foraging trips and night time sleeping patterns. Moreover, ethnographic studies have shown that co-residing family units engage in the most frequent daily interactions (Jones et al. 2009). We would also like to refer to other studies that have used individual encounters during daily tasks (Hamilton 2021) or overlapping use of space (Pearce 2014) as estimators of the extent of social connections and grouping.

In general, I think it would be helpful if you provide more explanations for particular motivations behind the choices you made to set up the model. You sometimes have details in your response to the reviewers' comments that would be helpful to include in the manuscript.

For example, in your reply, you refer how you built on the model by Ramos-Fernández and colleagues, as well as Binford/etc. However, the manuscript does not describe what these earlier studies found (right now you only describe what they did).

We have now highlighted the part where we had briefly described the findings by Ramos-Fernandez et al 2007 in the first paragraph of the methods section, and added an important and relevant finding from Binford's ethnographic studies.

In addition, you do not mention explicitly what you are changing (distribution/richness of environment; whether individuals move constantly through the environment or whether they move to fixed locations from which they exploit the local environment before moving to the next location).

We mention the key modification that we made in the model which was to modify the resource-maximization rule to implement central-place foraging (in Methods, line 87). We have now added another sentence to better explain the change we made.

In your introduction you never explicitly mention what the outcome is you are expecting to see in your model (social network that emerges based on the encounters when two or more individuals forage in the same patch).

We now mention this more clearly in the Introduction (paragraph starting on line 54). However, the details are present in the Model description (line 83).

In the reply you argue for why richness and heterogeneity are linked, but you don't explain that in the manuscript.

We now mention this more clearly. See lines 76-82 as well as Figure 1b where we provide examples of how the environment looks like at different values of beta.

You have added the simpler network measures to help to understand whether the emerging social interactions are realistic and likely to reflect those found in hunter-gatherer societies. While you state that the networks you find resemble those described from actual interactions, it would be helpful if you could specify the actual details (e.g. what is the range of values reported for a particular measure in the literature).

We have now added references to ethnographic studies from real hunter-gatherer societies around the world, and explained how our network measures relate to data from such studies. See paragraph starting on line 302 and lines 105-111.

It was also not clear to me why you choose population size of 50, 100 and 200? How does this relate to number of individuals across societies. To me, this appears to be the number of a given community that then splits with individuals foraging individual out of these camps (e.g. camps of Hadza regularly consist of 50 individuals). As such, you are simulating the behaviour of individuals within one single group, and I am not sure you are getting at the multi-level aspect that you focus on in the introduction and discussion.

The main purpose of running the model with different number of agents, as raised by reviewers, was to theoretically explore the effect of different population sizes on our model results. Whilst there are good estimates of average band sizes of HG societies across the world, there is a lack of research on the upper bound of hunter-gatherer social groupings and hence a lack of consensus on the size of regional bands or ethnolinguistic units. We mostly focus on population sizes of 100 foraging units/families (or 500 individuals) which have been documented widely and called the “magic number” estimated by Steward (1968) and assumed to represent the average size of hunter-gatherer regional communities or groups (Kelly, 2013). Moreover, populations of size 200 correspond to the size of entire ethnic populations (or metapopulation sizes) (as compiled in Lehmann et al. 2014). On the other hand, populations of 50 (200-250 individuals) correspond to estimates of the lower limit of viable hunter-gatherer populations by Wobst (1974). We have now added our reasoning in the Model Description (Line 305)

I also had another comment about the population size analyses: while you spent a whole paragraph of the discussion on the effect of population size, this is mentioned in a single, non-quantitative sentence in the results. If you consider something a key finding, please present the analytical outcome in the results.

We thank the editor for suggesting this, we have now added the effect of population sizes to our results (see Figure 5 and section 3.3) as well as our abstract.

Please let us know if there is anything else you would like us to revise. We look very much forward to your decision.

Sincerely,

Cecilia Padilla-Iglesias, Ketika Garg, Nicolás Restrepo Ochoa and Bleu Knight

Below is the response to the editor and reviewers of Proceedings of the Royal Society B.

Dear Editor,

First of all, we would like to thank the reviewers for the detailed comments and for giving us the opportunity to revise our manuscript. We believe we have addressed all questions raised by the reviewers and hope you are satisfied with the replies as well as the new manuscript. In general, we have expanded on the rationale behind the structure of the model and how it explicitly relates to human foraging strategies and the pre-existing literature on them (such as Binford's 1980 logistic vs. residential mobility terminology). We have now provided additional information on the utilised network metrics as well as included the additional measurements suggested by reviewer 1, and reported the evolution of such metrics and their impact on information transmission (as recommended by reviewer 3). We have also performed sensitivity analyses to assess the effect of the duration of the simulation on the model outcomes that show the robustness of our findings. Below you can see a detailed account of the changes made as well as our response to the specific comments made by the Associate Editor and each of the reviewers.

Associate Editor Comments to Author:

It is currently unclear why the agents are individuals and how this fits to human foraging strategies or/and how this affects the model outcome.

We thank the reviewer for their remarks. We wish to clarify that agents are individuals because the purpose of this model is to assess how individual-level simple foraging decisions guided purely by ecology can lead to grouping patterns and social networks without the need of having explicit social rules. That is, we are not trying to deny the effect of social factors themselves on social organization, but we are saying that even when *only* considering (and comparing) the foraging patterns of individual hunter-gatherers and those of other primates (including great apes), we can already observe emergent social networks that are particularly efficient at transmitting information. As we mention in the last paragraph of the discussion (line 255 onwards), once a system is in place where information flows efficiently, a self-reinforcing system can kick-off whereby more efficient information transmission leads to more structured social networks that further increase that ability, for example, collective foraging.

Reviewer(s)' Comments to Author: Referee: 1

Comments to the Author(s)

1. Foragers live in groups

As the authors say in the discussion there are all sort of complexities of forager social organisation that could be added to the model. In general, I agree that we should aim for simple models and not add complexity for the sake of it. Nonetheless, this model misses one critical aspect of forager social organisation: the fact that individuals live in groups. Central-place foraging is about groups of individuals setting up camp together and then individuals going off foraging alone or in small sub-groups. This is fundamental and ignoring it undermines the meaningfulness of the current model results, in my opinion.

What could be done about this? One option would be to reconceptualise each agent in the model as a group. Then the network could be considered an inter-camp interaction network. But if the authors were to do this they would still want to account for the migration of individuals between camps and for the fact that once a group establishes a home base, individuals then can go off foraging alone or in sub- groups and that their social interactions can occur both at their foraging site and back home at the home base. This would require significant additions to the model.

We thank the reviewer very much for their detailed feedback. The baseline model to which we add the "central place" dimension simulates individual foragers moving across the landscape simply to optimise their foraging returns. Our manuscript compares that baseline model with a model where foragers not only maximise their foraging returns but *also* return to and change central places. We do not ignore that foragers live in groups, in fact, our model investigates whether the formation of such groups and some of the derived properties that we observe in them could partly be the result of individual foraging-related decisions.

We are aware that the location of central places in real life is by no means exclusively dependent on ecology, and that other social factors may further shape social networks. Nonetheless, our model does not wish to create an exhaustive representation of the

complex and diverse factors that may affect hunter-gatherer social organization but to tease apart the implications of ecology on derived features of hunter-gatherer foraging-related mobility for their social networks and their ability to transmit information. In other words, whilst social motivations might reinforce structured human social networks - we wanted to determine whether hunter-gatherer foraging-related mobility *alone* could have served as a precursor to the complex social organization observed in real hunter-gatherer societies by means of creating a structure efficient at information transmission, that would have subsequently facilitated any further structuring.

In addition, whilst collective foraging is certainly practiced, individual foraging (yet returning to a central place) is also quite prevalent among contemporary hunter-gatherers, and particularly for gathering, very often is done individually (See Lee, 1979 or Marlowe, 2010 for extensive ethnographic accounts of hunter-gatherer foraging-related mobility for both hunting and gathering or see Wood et al. 2021 for a detailed account of individual mobility patterns in Hadza men and women). Hence, in order to disentangle the effect of central places from that of collective foraging on hunter-gatherer social organization, we deemed it important to first consider the effect of central places *alone* before embarking on further investigations about adding on social organizations usually associated with central-place foraging. In other words, our model shows that even without collective foraging/pre-existent social organization, the simple addition of home bases, could have already pre-adapted our ancestors' social networks for greater information transmission. We have also added this explanation now to our discussion (line 252 onwards).

2. Network efficiency

Network efficiency is a rather abstract measure. I'm not suggesting that it should be taken out but I think that it would be illuminating to provide the simpler measures that change with central-place foraging and which influence network structure. For example, what does central place foraging do to the number of interactions per turn and the number of unique interactions in the 100 trial period?

We have now added figures with the number of total interactions per time-step, number of unique interactions per time-step and the number of components in the network (see supplementary figures 6-13) throughout the simulation, so that the reader can see how these vary as the simulation proceeds. In the main and supplementary methods sections text, we have also further explained the effect of the number of interactions and network components on the efficiency metrics, as well as the metrics themselves.

3. Patch depletion

What determines the rate of patch depletion? Are patches replenished after depletion?

The patches in our model are not replenished but following the reason and model design by Ramos-Fernández et al (2006), the number of patches is sufficiently high to not have a significant effect on the results obtained. Patches deplete whenever a forager is present at a patch, and hence, the rate of patch depletion is linked to the heterogeneity of the environment and the foraging radius of the foragers. When the environment is abundant (Beta = 1.5), foragers do not move or coincide much causing the patches to deplete in a

linear fashion. However, when the environment is heterogeneous and some patches are richer than others ($\text{Beta} > 1.5$), foragers coincide at those rich patches more frequently and the patches deplete in a non-linear fashion. These patterns are congruent with realistic trends, whereby resource clustering leads to agents coinciding in the same patches and thus making them deplete disproportionately faster than the surrounding ones. We now provide plots of patch depletion rate throughout the simulations in our supplementary materials (Supplementary figure 5) and more explanation in the Supplementary Results.

Referee: 2

Comments to the Author(s)

I enjoyed reading this paper. It was interesting and well written and I have no criticisms or comments of substance. I was a bit struck, however, by the strident tone of lines 55-58:

"Previous computational models have explored the effects of environmental heterogeneity on social networks emerging from foraging behavior across different environments (35) and hunter-gatherer mobility on cultural transmission (10; 36). However, an explicit link between foraging strategies, environmental features, and hunter-gatherer interaction networks has not been made yet."

To the mind of this reviewer, all of human behavioral ecology is focused on these explicit links. It is an overriding theme of the discipline even if a computational model is lacking. I would suggest tempering these words to say something about using a computational model to test or corroborate known or presumed links.

We thank the reviewer and editor for this suggestion, we have now changed our sentence to mention the lack of computational models as opposed to research in general (see last paragraph of the Introduction, starting in line 52). However, we sustain that quantitative approaches to the study of a potential causal relationship between foraging strategies, environmental features and hunter-gatherer interaction networks remain indeed scarce and thus we believe the timeliness and contribution of this model to be substantial.

Referee: 3

Comments to the Author(s)

To begin with a general point, there are of course many species that practice central-place foraging of various kinds; it would therefore be useful to broaden the scope of the paper in the discussion. Since the results rely purely on simulation, can they be applied to other species? If not, how might the models be modified to allow this (or, on a related point, do the authors believe there is anything unique about human central-place foraging)? Expanding the paper in this way – even if it's just a flag in the introduction and a paragraph in the discussion – might make the paper more appealing to the wider readership of PRSB.

We agree with the reviewer that some other species practice central-place foraging, however, the focus of the paper is to study the effect of some of the derived features of

central place foraging on our lineage. That is, we wished to study the implications of central-place foraging on an already cognitively sophisticated, communicative Great Ape with an ability for social and cultural learning.

Hence, whilst convergence is an interesting phenomenon in biology, here we are interested in the derived features of human social organization and hence, our model stems from a chimpanzee-like baseline to which features characteristic of our lineage are included and compared. If the model were to be adjusted to model other central-place foragers like birds and social insects, care should be placed on the choice of “baseline” for comparisons, as our chosen baseline may no longer represent the background onto which central place foraging may have been introduced.

To simulate the central-place foraging in other animals, we would also need to modify the foraging decisions and rules. For example, our model closely simulates Kelly and Binford’s model for human hunter-gatherer foraging and residential mobility, which would not be suitable for say, ants. However, it would certainly be interesting to compare central-place foraging and its effects across species.

There are several issues related to the modelling that could be better grounded in the ethnographic record, and merit some discussion in the text:

1) Why are the agents individuals rather than foraging groups (i.e. subgroups of the wider ‘band’ pattern of hunter-gatherer social organisation)? While we appreciate that interactions may take place between individuals, this may not be a particularly realistic picture of human foraging practices. It would be interesting to at least discuss how this would affect the simulations (it may be that it would affect the results very little). In many cases bands consist of groups of individuals that each have different extra- band links, and this overall high degree of connectivity is a great strength. So the number of links that a band possesses might be far more important than the number of links that an individual possesses, and the number of individual links between bands could then be an index of the strength of connectivity (or the weights in the modelling).

We completely agree that collective foraging can add an additional layer of structuring, especially at a regional level. Please see our response concerning this issue to Reviewer 1’s first point as well as the newly included discussion paragraph starting on line 270 , where we discuss the importance of disentangling the effect of social networks of environmentally-driven influences on individual foraging decisions from those of socially-driven motivations to forage with others whilst keeping in mind that the latter may add a further layer of structuring to hunter-gatherer social networks.

2) Why doesn’t logistical mobility (in which there are longer foraging trips that stray further away from, but still ultimately return to, the home base) factor into this model? This is a mode of subsistence mobility that is very common in human foragers, and could be factored in via the method described by Perreault and Brantingham (2011) – a study that the authors cite. Logistical mobility would effectively extend the foraging radius and reduce the frequency of residential moves, which could have important effects on the results.

Our model does simulate logistical mobility separately from residential mobility, where foragers conduct logistical forays within a given radius and move their residence when resources deplete within the foraging radius. However, we labelled logistical moves as 'foraging' moves in the paper. We have now stated clearly in the text that 'foraging' moves are the same as 'logistical' moves.

3) The movements of central places are basically following Binford's 'double radius leapfrog pattern', so you can explicitly tie it to his research into hunter-gatherer mobility. He also discusses 'point-to-point' foraging.

We had referenced Binford's original texts when defining the terms, but now we have also explicitly mentioned the equivalence in terminology.

4) It is not clear how / whether patches are depleted or regenerate. Do foragers deplete patches by some given amount per unit time? Do patches regenerate according to a logistic model? These are commonly used features of previous models, but there is no information given on these aspects in the paper.

The baseline model to which we compare our model does not include patch regeneration, hence, our model (for comparability purposes) doesn't include it either. For a more in-depth explanation of why this is the case, please see our response to point 3 raised by Reviewer 1, as well as the modified text (see line 76-77), where we clearly state that patches do not regenerate. In addition, please refer to Figure S5 where we have now included plots of the rate of resource depletion throughout the simulation across the different environments.

5) How are foragers seeded into the landscape in the first place? Do you drop a given number of individual foragers into random cells? Or are they aggregated in some way into bands? This is important to report in the paper, as it affects how realistic the results are.

The foragers are randomly and independently distributed on available patches. We have now made this clear in the model description. See line no. 78

6) Remember that Ramos-Fernandez and colleagues were testing a particular hypothesis, that power-law distributed resources could lead to power-law distributed movement patterns (back in the days of fascination with Levy flights). The fission-fusion result they achieved was just the product of a kind of dynamic ideal free distribution. The elegance of their model lay in the fact that complex dynamics could emerge from very simple rules that did not assume anything about social relationships (foragers associated with one another simply because they were simultaneously drawn to rich patches). The authors need to make it clear how their results differ from those of Ramos-Fernandez and colleagues; there is an additional assumption that social links are established when foragers meet, and that these persist (see below). The results of Ramos-Fernandez and colleagues suggested that longer moves would be more likely with higher beta values, and that should be sufficient to create the kind of 'small world' networks observed here, without the need for central place foraging to create this effect.

In our paper, we have compared the networks that resulted from the original Ramos-Fernandez et al.'s paper with the ones from our model now with central-place foragers, but the way in which links are made is exactly the same in our as in the paper by Ramos-Fernandez et al. In other words, in both models, social network (or interaction) links are established when two foragers meet in the same location. Similarly, in none of the models are social factors considered (or social relationships) in conditioning individuals positional or mobility decisions. We label the original networks and models as 'point-to-point' foraging based on the previously established terminology. The main result from the original model showed that at intermediate values of beta (or an environment where rich patches are not too scarce and not too abundant), the networks formed depict a complex social structure with many cliquish, sub-groups. Our analyses take this finding further and suggest that these networks have high local efficiency (or cliquishness) with many well-connected sub-groups which do not have connections between them. However, when we ran the model with central-place foraging, we found that at intermediate radius, the networks formed not only have a high local efficiency but also a higher global efficiency (which resembles small-world characteristics).

7) The language used to describe the environment needs to be revised a little; smaller values of beta do lead to more homogeneous environments, but with large values of beta there are a few very rich patches and lots of very poor patches. So rather than beta = 1 indicating 'many rich resource patches' it would be better to stick to saying there is greater homogeneity, because there probably won't be any patches in a beta=1 environment that are as rich as the richest patches in a beta = 3 environment, for example. Also, beta = 4.5 is not a 'resource poor environment', it's just highly heterogeneous. It's not clear whether the authors actually vary k and beta independently, or how exactly the power law distribution of resources is normalized.

We wish to clarify that the lower values of beta will result in more heterogeneous environments while higher values, like 4.5 will result in greater homogeneity. Following the Ramos model, Beta and K co-vary. Given that a lower value of beta can result in a K of values really high, the environment will be overall abundant in resources as well (see Figure S5 for initial values of resource content in a particular environment). Similarly, when beta is high, for example, 4.5, K is restricted to a smaller maximum value resulting in an environment that is not as rich in resources. We have rephrased our explanation of the environment and added the normalization constant in the Methods section (see paragraph starting on line 69).

8) How realistic is it to have a randomly heterogeneous environment in terms of patches – surely there would be an element of spatial autocorrelation to patch richness, and this would in turn have a substantial impact on the spatial pattern of contacts between foragers?

In our model, the way that the beta parameter operates (by linking abundance and heterogeneity) does result in environments that can be considered to be spatially autocorrelated to an extent, atleast at a coarse-grained level. For instance, when the beta is either very low (1.5) or high (4.5), the resources are spatially uniform (i.e, either almost every patch is rich or not). But when beta is intermediate, the environment is more fragmented where food is clumped into different parts of the environment. We believe that

this simulates environments similar to real-life environments. For example, less abundant and uniform environments (with high beta value) can represent more arid environments with low productivity. Whereas, lower beta values can represent environments that are high in productivity, for example, the tropics.

More specifically, our model's environment is similar to the environments outlined in Binford (1980) and Kelly (1983, 1995) where environments are described by variance in resources and the net productivity (an approach also adopted in other modelling frameworks such as that by Premo, 2015 or Premo and Torstevin, 2016). Moreover, we wished to compare the baseline model (from Ramos-Fernandez) as well as specifically assess how environmental fragmentation and productivity affected foraging related mobility (as we motivate in our Introduction) and thus, the networks.

9) Lines 139-40: foragers primarily made short residential moves when the radius was small. But surely this is not a finding, it's built into the model – they can only make residential moves of 2 the radius...?*

We have rephrased this statement now (see line 149-152). However, we wish to clarify that we do not impose the distance at which residential moves are made but a *minimum* distance they should have (see lines 89-90). Again, a similar approach was taken by Premo (2015) or Binford (1980), simply to avoid overlap between home bases. We have now also cited Premo's modeling decision in our text.

10) Similarly, in lines 213-4: social networks formed with larger radii are more expansive – you don't need a model to show this surely? It would be better to re-phrase this in terms of your specific results.

We have rephrased these lines now (see 227-229) in terms of our specific results. However, we wish to clarify that our intention is to first describe our results in a general and overarching manner to connect to the previous literature about network expansions. We get into the more specific results in the following paragraphs.

11) Perhaps the most important issue relates to the transition from the modelling results to the network analyses. The networks are assumed to be static in the cultural transmission analyses, but are in fact dynamic; the flip-side of this is that the size and structure of the networks depend upon the number of iterations that the foraging simulation is run for. The weighting of the networks helps to deal with this to an extent, but we would suggest that some network metrics are collected during foraging simulations to show (or to see whether) these metrics stabilise (asymptote) during simulations. Otherwise, the 1,000 iteration cut off is arbitrary and could have a substantial effect on the results. Alternatively, just extract networks every, say, 100 iterations of the simulation and check to see whether your measures of efficiency correlate in any way with the iteration number. Ideally your efficiency measures would stabilise at some point, otherwise the results are highly dependent on the number of iterations and can't be considered as reliable reflections of the foraging systems. This issue will have to be addressed prior to publication.

If networks don't stabilise, the authors should consider incorporating some measure of the extent to which they change through time; this could be an interesting expansion, and would be informative in terms of how networks form (and decay).

This is a crucial point and we thank this reviewer for bringing it up. In order to show that our results are not an artifact of an arbitrary cut-off or of the aggregation of all interactions, we took the following steps. First, we ran a sensitivity analysis comprising 20 runs, of 1000 time-steps each, for all parameter combinations. We then analyzed the cumulative development of the networks, examining the resulting graph after a certain interval of time-steps. We show that our results are robust and do not vary substantially based on how long the simulations are run for. More specifically, we found that our efficiency measures mostly stabilize over time. We have added our new findings in the supplementary text. These analyses certainly corroborate the robusticity of our findings, and we are very grateful to have been suggested to include them.

Miscellaneous:

Given the vexed history of the term in anthropology, it might be wise to remove 'home base' from the paper, and refer exclusively to 'central places'; this also facilitates comparison with other species.

We have now replaced all mentions of home-bases with homes, home location or central places according to what's most appropriate in each instance.

The authors might consider using the ODD Protocol for describing ABMs detailed by Grimm and colleagues. This forces you to make all aspects of the modelling very clear, including scheduling other important features. You could retain a basic verbal description in the main text but include a full ODD description in the supplementary materials; this also makes it easier for people to replicate or expand upon your model using different programming languages / development environments.

We agree that a detailed description of the model is essential in order to ensure replicability of the findings and future usage of the model and have created an ODD protocol that we have attached as supplementary materials. In addition, we have created a Github repository where not only will the code to reproduce the model is available but in addition a detailed description of the characteristics of each element of the model (agents and environments), as well as the ODD description. It also contains the necessary code to reproduce all the analyses performed on the model output and the data obtained across all our simulations.

Please let us know if there is anything else you would like us to revise. We look forward to your decision.

Best,

Cecilia Padilla-Iglesias, Ketika Garg, Nicolas Restrepo Ochoa and V. Bleu Knight